# Frame Mining: a Free Lunch for Learning Robotic Manipulation from 3D Point Clouds

**Minghua Liu[1*], Xuanlin Li[1*], Zhan Ling[1*], Yangyan Li[2], Hao Su[1]**
[1]UC San Diego  [2]Alibaba
https://colin97.github.io/FrameMining/

**Abstract:** We study how choices of input point cloud coordinate frames impact learning of manipulation skills from 3D point clouds. There exist a variety of coordinate frame choices to normalize captured robot-object-interaction point clouds. We find that different frames have a profound effect on agent learning performance, and the trend is similar across 3D backbone networks. In particular, the end-effector frame and the target-part frame achieve higher training efficiency than the commonly used world frame and robot-base frame in many tasks, intuitively because they provide helpful alignments among point clouds across time steps and thus can simplify visual module learning. Moreover, the well-performing frames vary across tasks, and some tasks may benefit from multiple frame candidates. We thus propose FrameMiners to adaptively select candidate frames and fuse their merits in a task-agnostic manner. Experimentally, FrameMiners achieves on-par or significantly higher performance than the best single-frame version on five fully physical manipulation tasks adapted from ManiSkill and OCRTOC. Without changing existing camera placements or adding extra cameras, point cloud frame mining can serve as a free lunch to improve 3D manipulation learning.

**Keywords:** point cloud, coordinate frame, robot manipulation, 3D, RL

## 1 Introduction

With the rapid development and proliferation of low-cost 3D sensors, point clouds have become more accessible and affordable in robotics tasks [1]. Also, the tremendous progress in building neural networks with 3D point clouds [2, 3, 4, 5, 6, 7] has enabled powerful and flexible frameworks for 3D visual understanding tasks such as 3D object detection [5, 8, 9], 6D pose estimation [10, 11], and instance segmentation [12, 13]. Very recently, point cloud started to be used as the input to deep reinforcement learning (RL) for object manipulation [14, 15, 16], which aims at learning mappings directly from raw 3D sensor observations of unstructured environments to robot action commands. These end-to-end learning methods avoid highly structured pipelines and laborious human engineering required by conventional robot manipulation systems.

When building an agent with point cloud input, existing works [14, 15, 16] typically incorporate off-the-shelf point cloud backbone networks (e.g., PointNet [2]) into the pipeline as a feature extractor of the scene. However, some facets in constructing point cloud representations have been overlooked. For example, in the literature of 3D deep learning, the choice of coordinate frame significantly affects

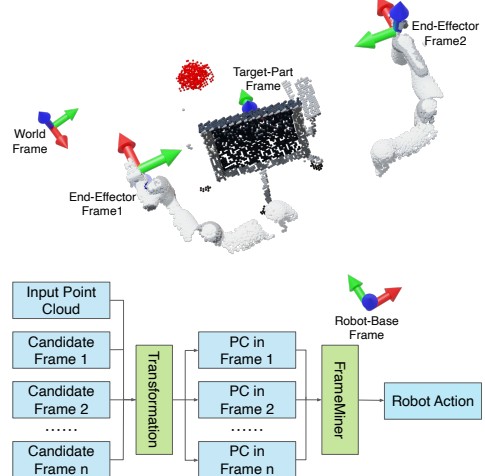

Figure 1: A 3D point cloud of a dual-arm robot pushing a chair, which can be represented in various coordinate frames without changing camera placements or requiring extra camera views. Our FrameMiner takes as input a point cloud represented in multiple candidate frames and adaptively fuses their merits, resulting in better performance.

---

*equal contribution

6th Conference on Robot Learning (CoRL 2022), Auckland, New Zealand.

task performance [2, 17, 18, 19, 20, 21]. On 3D instance segmentation benchmarks for autonomous driving, previous work such as [5] showed a pipeline to process input point clouds in the camera frame, frustum frame, and object frame subsequently, leading to a large performance boost in comparison to using the camera frame alone. For our goal of manipulation skill learning, point clouds describe dynamic interactions between robots and objects, including frequent contacts and occlusions. This is a novel and more complex setting that differs from well-explored scenarios in 3D supervised learning (e.g., single objects, outdoor scenes for autonomous driving). Under this setting, choices of coordinate frames are more flexible and diverse as multiple entities (e.g., robot and manipulated object) and dynamic movements are involved.

In this work, we first examine whether and how different coordinate frames may impact the performance and sample efficiency of point cloud-based RL for object manipulation tasks. We study four candidate coordinate frames: world frame, robot-base frame, end-effector frame, and target-part frame. These frames differ in positions of origin and orientations of axes, and canonicalize inputs in different manners (e.g., a fixed third-view, ego-centric, hand-centric, object-centric). The comparison and analysis are performed on five distinct physical manipulation tasks adapted from ManiSkill [22] and OCRTOC [23], covering various numbers of arms, robot mobilities, and camera settings. Results show that the choice of frames has profound effects. In particular, the end-effector frame and the target-part frames, rarely considered in previous works, lead to significantly better sample efficiency and final convergence than the widely used world frame and robot-base frame on many tasks. Visualization and analysis indicate that, by using different coordinate frames to represent input point clouds, we are actually performing various alignments of input scenes through $\mathbb{SE}(3)$ transformations, which may simplify the learning of visual modules.

However, the well-performing single coordinate frame may vary from task to task, and in many cases, we may need coordination between decisions made according to multiple coordinate frames. For example, tasks equipped with dual-arm robots may benefit from both left-hand and right-hand frames. For mobile manipulation tasks involving both navigation and manipulation, different frames could favor different skills (e.g., robot-base frame for navigation skills, end-effector frame for manipulation skills). We thus propose three task-agnostic strategies to adaptively select from multiple candidate coordinate frames and fuse their merits, leading to more efficient and effective object manipulation policy learning. Because we do not need to capture additional camera views or rely on task-specific frame selections, our frame mining strategies can be used as a free lunch to improve existing methods on point cloud-based policy learning. We call these fusion approaches as *FrameMiners*. Experimentally, we find that it matters to fuse information from multiple frames, but the specific FrameMiner to choose does not create much performance difference. In particular, we use one of the FrameMiners, MixAction, to interpret the importance of different frames in the policy execution process, and the interpretation agrees with our intuitions.

In summary, the main contributions of this work are as follows:

- We find that the choice of coordinate frame has a profound impact on point cloud-based object manipulation learning. In particular, the end-effector frame and the target-part frame lead to much better sample efficiency than the widely-used world frame and robot-base frame on many tasks;

- We find that well-performing frames differ task by task, necessitating task-agnostic ways to select and fuse frames. This observation is consistent across 3D backbone networks;

- We propose FrameMiners, a collection of methods to fuse information from multiple candidate frames. FrameMiners provide a free lunch to improve existing point cloud-based manipulation learning methods without changing camera placements or requiring additional camera views.

## 2   Related Work

**Manipulation Learning with Point Clouds**   Visual representation learning for object manipulation has been extensively studied [24, 25, 26, 27, 28, 29, 30, 31]. With the flourishment of 3D deep learning [2, 3, 4, 5, 6, 7], a major line of work learns representations from 3D point clouds for object manipulation [32, 33, 34, 35, 36, 37, 38]. Recently, people have also started to incorporate point clouds into deep reinforcement learning (RL) pipelines for manipulation learning [14, 15, 16]. However, existing point cloud-based manipulation learning methods have not paid enough attention to coordinate frame selections of input point clouds, which is fundamental in 3D visual learning. Some very recent work [39, 40] explored placement and selection of camera views and fusion of multi-view images. We differ from them in that we focus on the preprocessing of captured input point clouds without modifying existing camera configurations or adding additional cameras.

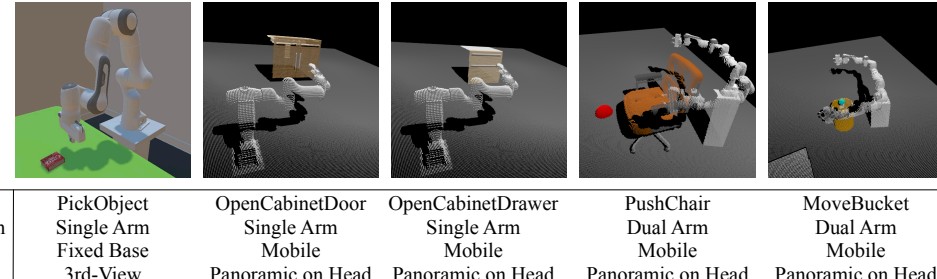

| Task | PickObject | OpenCabinetDoor | OpenCabinetDrawer | PushChair | MoveBucket |
|---|---|---|---|---|---|
| Robot Arm | Single Arm | Single Arm | Single Arm | Dual Arm | Dual Arm |
| Mobility | Fixed Base | Mobile | Mobile | Mobile | Mobile |
| Cameras | 3rd-View | Panoramic on Head | Panoramic on Head | Panoramic on Head | Panoramic on Head |

Figure 2: We study coordinate frame mining on manipulation tasks adapted from OCRTOC [23] and ManiSkill [22] covering various setups (e.g., #arms, mobility, camera). Simulation is fully physical.

**Normalization and View Fusion in Point Cloud Learning**  Normalizing input point clouds is a common practice in 3D deep learning literature. For example, in single object analysis (e.g., classification and part segmentation), people often normalize input point clouds into a categorical canonical pose with unit scale [2, 3], simplifying network training. Prior works find that existing point cloud networks [2, 3, 7, 41, 42] are very sensitive to input normalization [21, 43, 44], and many recent attempts explore rotation invariant [45, 46, 47] and equivariant methods [21, 48, 49] for 3D deep learning. Compared to well-studied scenarios (e.g., single object and autonomous driving), normalizations of point clouds under robot-object interactions are under-explored.

In LiDAR point cloud learning for autonomous driving, many work focuses on the fusion of multiple views [17, 18, 19, 20]. Unlike fusing multiple camera scans, there is only one point cloud. They propose to process the point cloud from different views (e.g., perspective view and birds-eye view) to combine their merits, which has proven to be helpful. Our work shares a similar idea. However, we focus on robotic object manipulation settings, and the choice of coordinate systems is more diverse.

## 3 Point Cloud Coordinate Frame Selection Matters

### 3.1 Problem Setup

We aim to learn agents with point cloud input for object manipulation tasks via Reinforcement / Imitation Learning (RL/IL). A task is formally defined as a Partially-Observable Markov Decision Process (POMDP), which is represented by a tuple $M = (S, A, \mu, T, R, \gamma, \Omega, O)$. Here $S$ and $A$ are the environment

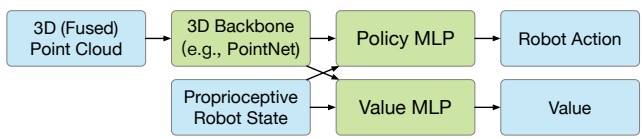

Figure 3: Architecture of a 3D point cloud-based agent, which is optimized by actor-critic RL algorithms. We study coordinate frame selection of input (fused) point cloud.

state space and the action space. $\mu(s)$, $T(s'|s, a)$, $R(s, a)$, and $\gamma$ are the initial state distribution, state transition probability, reward function, and discount factor, respectively. $O(s) : S \to \Omega$ is the observation function that maps environment states to the observation space $\Omega$. Our agent is represented by a policy $\pi : \Omega \to A$, which aims to maximize the expected accumulated return given by $J(\pi \circ O) = E_{\mu, T, \pi}[\sum_{t=0}^{\infty} \gamma^t r(s_t, a_t)]$. Note that $\pi$ does not have access to the environment state $s$ and only has access to the observation $O(s)$. In this work, $O(s)$ consists of two parts: (1) a 3D point cloud captured by depth cameras; (2) proprioceptive states for the robot, such as joint positions and joint velocities. For the first part, if there are multiple cameras, we fuse all point clouds into a single one by transforming them into the same coordinate frame and concatenating the points together.

Fig. 3 shows the architecture of a 3D point cloud-based agent, which we use to discuss in this section. It first exploits a 3D backbone (e.g., PointNet [2]) to extract visual features from a 3D (fused) point cloud. The extracted features are then concatenated with proprioceptive robot states and fed into separate multi-layer perceptrons (MLP) for action and value prediction. The input (fused) point cloud can be represented in different coordinate frames before being fed into the 3D backbone network, and *the choice of coordinate frame is independent of camera views*. For example, a point cloud captured by a camera mounted on the robot's head can be transformed into the end-effector frame. In this work, we study how point cloud coordinate frames affect sample efficiency and final convergence of object manipulation learning. Unlike prior works [39, 40], we do not change robot camera configurations (e.g., camera placement, inclusion of additional cameras).

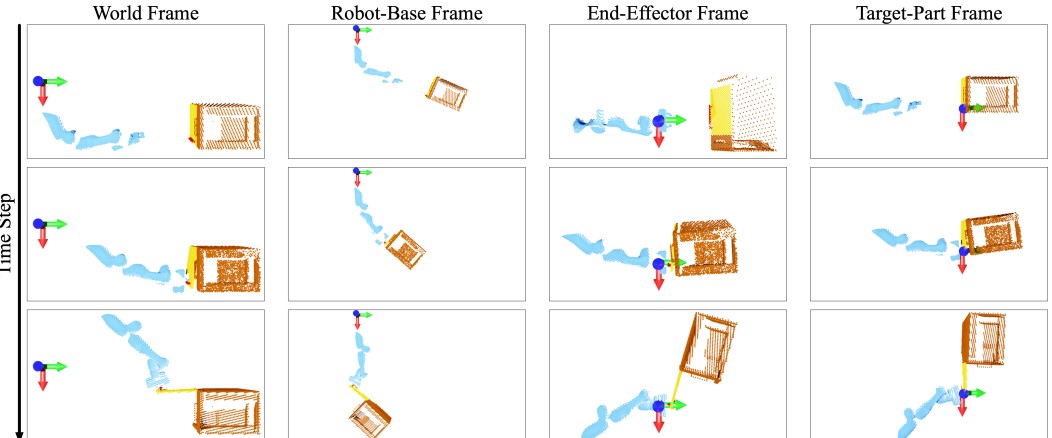

Figure 4: Illustration of four coordinate frames, which provide different alignments across time steps. We visualize three point clouds (three time steps) of an OpenCabinetDoor trajectory. Each row shows the same point cloud represented in different coordinate frames. Please zoom in for details. Robot arm, cabinet door handle, cabinet door, and cabinet body are colored in blue, red, yellow, and brown, respectively. RGB arrows indicate the corresponding origin and axes for each frame. Since the point clouds used for policy learning can be rather sparse, we show dense point clouds here for better visualization.

As shown in Fig. 2, we exemplify the frame selection problem on five fully-physical manipulation tasks, covering various numbers of robot arms, mobilities, and camera settings. Among them, PickObject is adapted from OCRTOC [23], and the other four tasks are adapted from ManiSkill [22]. On PickObject, a fixed-base single-arm robot learns to physically grasp an object from the table, lift it up to a target height, and keep it static for a while. Point clouds are captured from a 3rd-view camera. On ManiSkill tasks, agents learn generalizable physical manipulation skills (i.e., opening cabinet doors / drawers, pushing chairs / moving buckets to target positions) across objects with diverse topology, geometry, and appearance. We utilize mobile robots with one or two arms. Point clouds come from a panoramic camera mounted on the robot's head. Action space includes joint velocities of the arm(s) and the mobile robot base, along with joint positions of the gripper(s). More details are presented in the supplementary material.

### 3.2  Choices of Point Cloud Coordinate Frame

For 3D supervised learning tasks such as object classification and detection, it's a common practice to normalize input point clouds, and the choice of coordinate frames significantly affects task performance [2, 21, 17, 18, 19, 20]. In point cloud-based manipulation learning, we are faced with an underexplored, yet more challenging, setting. First, point clouds describe more complex robot-object interactions, possibly including frequent contacts and occlusions. Furthermore, compared to supervised learning, 3D visual modules receive weaker supervision signals during RL training. Therefore, it may become even more important to lessen the burden of visual module learning by properly normalizing input point clouds. Unlike previous well-studied point cloud learning scenarios (e.g., single-object point clouds, LiDAR point clouds for autonomous driving), there exist more diverse choices of coordinate frames. In this paper, we compare and analyze four candidates:

- A **world frame** is attached to a fixed point in the world (e.g., the start point of a trajectory).
- A **robot-base frame** is attached to the robot base, offering an egocentric perspective on a mobile robot. For a fixed-base robot, world frame and robot-base frame could be equivalent.
- In many object manipulation tasks, movements of robot end-effector(s) play important roles, and we can attach an **end-effector frame** to each of them. Note that for dual-arm robots, there are two end-effectors and thus two end-effector frames.
- A **target-part frame** is attached to the object part the robot intends to interact with (e.g., target door handle for the OpenCabinetDoor task).

When we transform captured point clouds into the world frame, the robot-base frame, and the end-effector frame, we may need proprioceptive robot states and potential robot movement tracking, which is typically accessible in modern robots. When we transform point clouds into the target-part

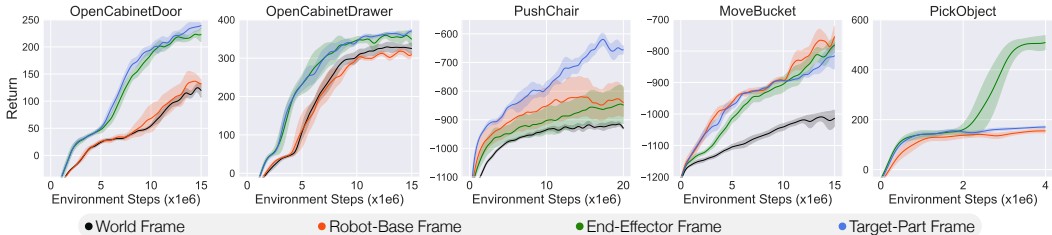

Figure 5: Comparison of four coordinate frames on five fully-physical manipulation tasks. The (fused) point cloud is transformed to a *single* coordinate frame before being fed to the visual backbone network. For dual-arm tasks (i.e., PushChair and MoveBucket), we use the right-hand frame as the end-effector frame. For PickObject, which has a fixed base, the world frame is the same as the robot-base frame. Mean and standard deviation over 5 seeds are shown.

frame, we may need to leverage off-the-shelf 3D object detection and pose estimation techniques. However, in this paper, we mainly focus on the choices of coordinate frames themselves. In our simulated experiments, we use ground truth object poses for the target-part frame.

In Fig. 4, we visualize an example trajectory under four coordinate frames. As shown in the figure, different coordinate frames canonicalize inputs in different manners (e.g., a fixed third-view, ego-centric, hand-centric, and object-centric), which is essentially performing various *alignments* among point clouds across multiple time steps. For example, in the end-effector frame, the end-effector is always aligned at the origin throughout a trajectory. Such alignments may simplify the learning of visual modules in distinct ways. With the end-effector frame, the network does not need to locate the end-effector in point clouds (always at the origin). Similarly, with the target-part frame, it can be easier to determine the relative position between the target part and the robot end-effector. The robot-base frame naturally aligns its frame axes with the moving directions of the robot's base.

### 3.3 Single-Frame Comparison on Manipulation Tasks

We compare the four coordinate frames on the five manipulation tasks by training PPO [50] agents using PointNet [2] as the 3D visual backbone. In this section, the (fused) point cloud is transformed into a **single** coordinate frame. For PushChair and MoveBucket tasks that use a dual-arm robot, we use the right hand frame as the end-effector frame (we observe almost identical performance using the left hand frame). For the target-part frame, we choose the handle frame for OpenCabinetDoor and OpenCabinetDrawer tasks, chair seat frame for the PushChair task, bucket for the MoveBucket task, and the target object for the PickObject task. Further details are presented in the supplementary.

Fig. 5 shows the results. We observe that distinct coordinate frames lead to very different agent training performances. Overall, the world frame is the least effective, especially in PushChair and Move-Bucket that involve more pronounced movement of the robot base. This suggests that the alignment of a static point in the world-frame is less helpful for the tasks. Compared to the commonly-used world-frame and robot-base frame, the end-effector frame has much higher sample efficiency on all single-arm tasks (i.e., OpenCabinetDoor, OpenCabinetDrawer, and PickObject), demonstrating the benefits of end-effector alignment. However, it shows similar or worse performance on PushChair and MoveBucket, intuitively because these tasks rely on dual-arm coordination, but our point cloud is normalized to a single end-effector frame (i.e., right hand frame). In addition, the target-part frame achieves the best sample efficiency on most tasks, suggesting that the target-part alignment across time could be of great help for point cloud-based visual manipulation learning.

### 3.4 Further Analysis

In robot manipulation tasks, agents often need to infer *binary relations* between subjects (e.g., relative pose between the end-effector and the cabinet handle). By aligning point clouds under certain frames (e.g., end-effector frame), these tasks may be reduced to *single-subject location tasks* (e.g., simply copying the handle pose), which become much easier to solve. To confirm this hypothesis, we perform a diagnosis experiment on OpenCabinetDoor, where we intentionally remove all robot points (i.e., blue points in Fig. 4) and see its effect on different coordinate frames. As shown in Fig. 6, after the robot points are removed, the end-effector frame performs the same, while the robot-base frame performs worse (the task is still solvable since the end-effector position is also provided in the proprioceptive robot state). This suggests that the end-effector frame allows an agent to focus on the target object, along with its interaction with the robot hand, which verifies our hypothesis.

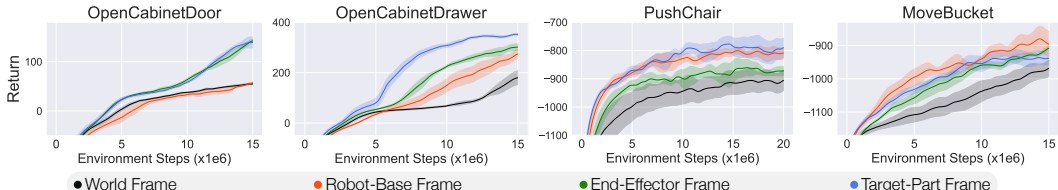

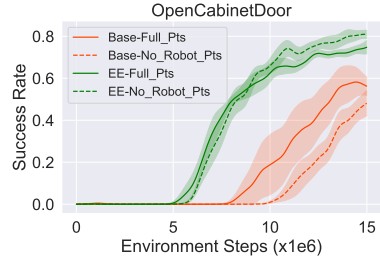

Figure 7: Using SparseConvNet [51] as the 3D visual backbone, we observe similar trends as Fig 5. Mean and standard deviation over 5 seeds are shown.

We utilized PointNet [2] as our 3D visual backbone for its fast speed and general good performance. However, it's unclear whether point cloud frame selection is also crucial for other 3D backbones, especially those more complex and powerful. Therefore, we conduct the same experiments as Sec 3.3 using SparseConvNet [51], a heavier 3D backbone network, on the four ManiSkill tasks (further details in the supplementary). As shown in Fig. 7, we observe similar relative performance between frames as before (e.g., the world frame performs poorly; the end-effector frame outperforms the world frame and the robot-base frame on OpenCabinetDoor and Open-CabinetDrawer). Interestingly, using SparseConvNet doesn't improve the overall performance over PointNet.

Figure 6: Removing robot points does not harm the performance of the EE frame on OpenCabinetDoor.

## 4 Mining Multiple Coordinate Frames

We have shown that different point cloud coordinate frames lead to distinct sample efficiencies and final performances in manipulation learning. However, a single frame can perform well on some tasks but poorly on others, and we wish to find good frames in a task-agnostic manner. Moreover, for complex manipulation tasks, a single frame could be insufficient, and synergistic coordination between multiple frames could provide unparalleled advantages. For example, when robots are equipped with multiple arms, each arm may have its preferred coordinate frame (e.g., left-hand frame and right-hand frame). In addition, in tasks that involve simultaneous navigation and manipulation (e.g., on PushChair and MoveBucket, an agent needs to move towards the target while manipulating chairs or buckets), different frames could benefit different skills (e.g., robot-base frame for navigation skills, and end-effector frame for manipulation skills). Therefore, it is of great help to propose a prior-agnostic method that can automatically select the best frame from multiple candidates or combine the merits of them. Again, we are not talking about fusing multiple camera views. Point clouds from multiple camera views are first fused together into a single point cloud, before being transformed to each coordinate frame.

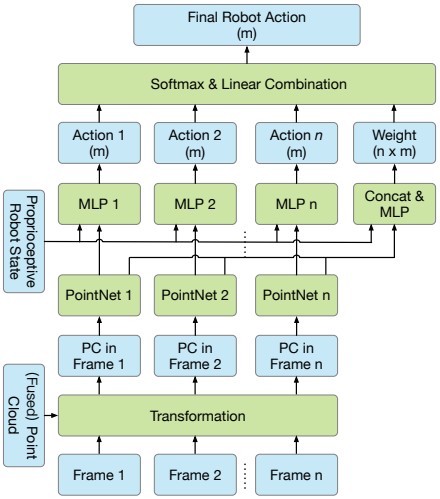

Figure 8: The pipeline of FrameMiner-MixAction (FM-MA). Each frame outputs an action proposal. Actions are then fused through input-dependent and joint-specific weights.

In this section, we will present a collection of three strategies to adaptively select and fuse multiple candidate coordinate frames, and we call them *FrameMiners*. In particular, we will first introduce *FrameMiner-MixAction* in Section 4.1 in detail to interpret the importance of different frames in the policy execution process. We will then briefly introduce the other two FrameMiners and compare different approaches with single-frame baselines.

### 4.1 FrameMiner-MixAction

Inspired by the idea of mixture of experts [52], we propose a general and interpretable framework, *FrameMiner-MixAction* (FM-MA). As shown in Fig. 8, FM-MA takes a (fused) point cloud and $n$

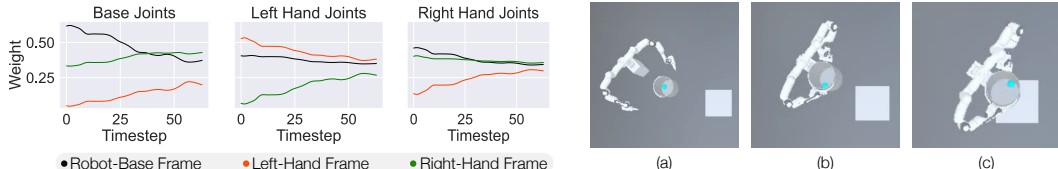

Figure 9: Left: learned weights of FrameMiner-MixAction over a MoveBucket trajectory, where three coordinate frames are fused. We divide robot joints into three groups and show the average weights of each group in each coordinate frame. Right: three stages of the trajectory. (a) Approaching the bucket. (b) Moving the bucket to the platform. (c) Placing the bucket on the platform.

candidate coordinate frames as input, and first transforms the point cloud into $n$ coordinate frames. For each transformed point cloud, FM-MA employs an expert network, consisting of a 3D visual backbone (e.g., PointNet [2]) and an MLP, to propose full robot actions (e.g. target velocity of $m$ joints). Since different experts use different coordinate frames, they are encouraged to specialize different skills and controls of different joints. Finally, we combine actions from the $n$ experts through input-dependent weights. Specifically, we concatenate extracted visual features from all $n$ frames with the proprioceptive robot state, feed it into an MLP, and predict a weight for each pair of expert and joint (there are $n \times m$ weights in total). For each joint, we normalize the weights over $n$ experts via softmax and fuse the actions through weighted linear combination.

FM-MA fuses actions by predicting joint-specific weights, since we believe that, for different joints, we need to extract information from different coordinate frames. Furthermore, the weights are input-dependent, potentially allowing the model to capture dynamic joint-frame relations at different stages of a task. Fig. 9 confirms these hypotheses. On MoveBucket trajectories, we observe distinct frame preferences between different robot joints. The left and right-hand frames contribute significantly to their respective joint actions. In addition, the weight distribution changes greatly over different trajectory stages. Initially, when the robot is moving towards the bucket but not interacting with it, the base frame contributes more. However, when the hands start to manipulate the bucket, the hand frames' weights increase. In particular, when the robot places the bucket onto the platform, we need careful coordination among all joints, and thus similar weights from each frame.

## 4.2   FrameMiners vs. Single Coordinate Frame

To study how network architectures influence coordinate frame fusion, we also propose other two strategies: *FrameMiner-FeatureConcat* (FM-FC) and *FrameMiner-TransformerGroup* (FM-TG). For each transformed point cloud, FM-FC uses an individual PointNet to extract visual feature. All visual features are then concatenated and fed into an MLP to predict robot action. FM-TG decomposes our robot action into three groups: base joint actions, left-hand joint actions, and right-hand joint actions (only two groups for single-arm tasks). After visual features are extracted from PointNets, they are fused through a Transformer [53] to produce a feature for each action group, which passes through an MLP to predict its respective joint actions (see the supplementary material for details).

We compare our three FrameMiners with single-frame baselines. *Specifically, in this section, we focus on frame mining among the robot-base frame and end-effector frame(s).* For the dual-arm tasks (i.e., PushChair and MoveBucket), the end-effector frames include both the left-hand frame and the right-hand frame. We will discuss the inclusion of target-part frame in Section 4.3. Fig. 10 and Tab. 1 show the comparison results. On single-arm tasks (i.e., OpenCabinetDoor/Drawer),

|        | Base   | EE    | FM-FC  | FM-MA   | FM-TG   |
|--------|--------|-------|--------|---------|---------|
| Door   | 54±7   | 80±2  | 79±3   | **84±2**| 70±6    |
| Drawer | 88±2   | 93±1  | **94±1**| 93±1   | 93±2    |
| Chair  | 7±3    | 2±1   | 32±4   | **36±4**| 34±6    |
| Bucket | 23±6   | 19±4  | 77±5   | 81±3    | **90±2**|

Table 1: Success rates (%) on four ManiSkill tasks.

our FrameMiners perform on par with the end-effector frame, which suggests that FrameMiners can automatically select the best single frame. On dual-arm tasks, our FrameMiners significantly outperform single-frame baselines, demonstrating the advantage of coordination between multiple coordinate frames. We also demonstrate that our FrameMiners outperform alternative designs and provide robust advantages (more details in the supplementary material). While it matters to fuse information from multiple frames, the specific FrameMiner to choose does not create much performance difference. Empirically, we find FM-MA less sensitive to training parameters, and FM-TG more computationally expensive.

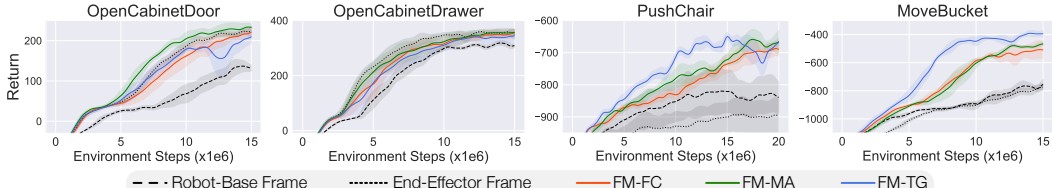

Figure 10: Comparison of different frame mining approaches on the four ManiSkill tasks, where the robot-base frame and end-effector frame(s) are fused. Black lines indicate single-frame baselines. Mean and standard deviation over 5 seeds are shown.

Our previous experiments were conducted using online RL. To investigate whether our previous findings can generalize to other algorithm domains, we perform experiments on imitation learning, and more details are presented in the supplementary material. The results corroborate our previous findings, i.e., different coordinate frames have a profound effect on point cloud-based manipulation skill learning, and FrameMiners are capable of automatically selecting the best coordinate frame or combining the merits from multiple frames and outperforming single-frame results.

### 4.3 Target-Part Frame

In Section 4.2, we focus on the fusion of robot-base frame and end-effector frames, since the target-part frame relies on pose estimation of target objects, which requires extra efforts in real-world settings. As shown in Fig. 11, if object poses are estimated, we sometimes observe further performance boost of FrameMiners from the target-part frame. For example, on PushChair, by incorporating the target-part frame, the success rate of FM-MA increases from $36\pm4\%$ to $53\pm3\%$. On PickObject, FM-MA already achieves good performance without the target-part frame ($94\pm2\%$); incorporating it slightly improves the success rate to $97\pm1\%$.

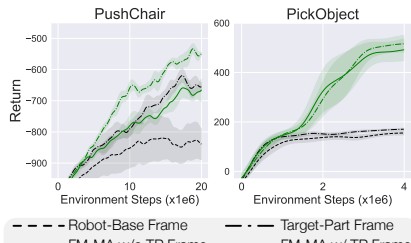

Figure 11: Fusion of target-part frame could further boost the performance.

## 5 Real World Experiments

To further verify that our learned policies can be deployed on real-world robots without introducing extra domain gaps, we test on Pick-Object with a Kinova Jaco2 Spherical 7-DoF robot, an Intel RealSense [54] D435 camera for uncolored point cloud capture, and a Rubik's cube from YCB objects [55, 56] (see Fig. 12). We use a 3-DoF end-effector position controller and a 1-DoF gripper position controller. We train the FM-MA policy (Section 4.1) by fusing the robot-base frame and the end-effector frame. At test time, we build a digital twin in the simulator over 25 sampled initializations of the real environment with a vision-based pipeline like Jiang et al. [57]. By following trajectories from policy rollout, we obtain a 84% success rate with FM-MA, compared to an 80% success rate with the end-effector frame. The robot-base frame is unable to achieve successful

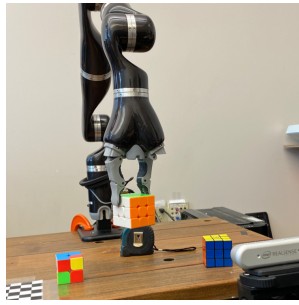

Figure 12: Real robot setup.

picks under our training budget. Note that the performance differences in the real world are very similar to the simulation environment, indicating that point cloud frame selection or mining does not affect original domain gap. More details are presented in the supplementary material.

## 6 Conclusion and Limitations

We find that choices of point cloud coordinate frames have a profound impact on learning manipulation skills. Our proposed FrameMiners can adaptively select and fuse multiple candidate frames, serving as a free lunch for 3D point cloud-based manipulation learning. Currently, our FrameMiners need to process each frame separately, leading to more network computation. In the future, we would like to explore more advanced fusion strategies to further improve network efficiency as well as performance. In addition, the target-part frame requires human judgment to determine the target part candidates and 6D pose estimation of the target parts, although we have shown our method can also achieve great improvements without the target-part frame (Section 4.2).

**Acknowledgments**

This work is supported in part by gifts from Qualcomm. We would like to thank Jiayuan Gu and Zhiao Huang for their helpful discussion and manuscript proofreading.

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
