# OpenReview forum: "Frame Mining: a Free Lunch for Learning Robotic Manipulation from 3D Point Clouds"
_robot-learning.org/CoRL/2022/Conference — CoRL 2022 Poster_

### Official Review · Reviewer_vXBE · 2022-06-30

**Originality:** Very Good
**Technical Quality:** Very Good
**Clarity Of Presentation:** Excellent
**Impact:** 4

**Recommendation:**

Strong Accept: I recommend accepting the paper and will argue for my recommendation even if other reviewers hold a different opinion.

**Summary:**

The paper identifies the importance of using particular coordinate reference frames for reinforcement learning with point-cloud inputs, demonstrates this experimentally, and proposes a novel technique to learn policies from a fusion of reference frames. Commonly used frames are investigated: world, robot base, end-effector and object frames (world and robot are meaningful to look at separately since a mobile robot is considered). Three different "frame mining" techniques are proposed (i.e. using information from different frames to select the action to execute), though it is shown in the experiments that while they do not differ significantly amongst themselves, they all provide improvements over using a single frame for the entire task. Experiments are performed on a variety of simulated platforms including a single fixed arm, a bimanual mobile robot, and a demonstration of the method on a real robot.

**Issues:**

Please see my comments in the strengths/weaknesses section. I would particularly appreciate the authors' input on the alternative to FM-MA I proposed (select max-weighted action), and the matter of a possibly moving camera that may be situated in an action-relevant frame (e.g. on the mobile base or on the end-effector, both common camera placements).

There are some smaller issues I identify here to improve the paper:
    - Figure 1, the 4th image caption should be "PushChair".
    - Pg. 3, last paragraph, second sentence, "... to extract visual features..." (pluralize features)
    - Figure 3, it would be easier to see with a notably different color for the door, yellow and orange are too similar.
    - Figure 8, the axes should be the same so that we can compare the plots on the same scale.

**Quality Of The Limitations Section:**

Additional details required

**Reviewer Expertise:**

5: The reviewer is absolutely certain that the evaluation is correct and very familiar with the relevant literature

**Robotics Focus:**

Sufficient demonstration on hardware

**Strengths And Weaknesses:**

The paper is very clearly written and in a more colloquial manner without sacrificing technical detail which I appreciate. I think the paper conclusions are intuitive and insightful. I believe the claims being made regarding the benefits of particular frames benefitting different tasks is known in robotics, but I have not seen a rigorous demonstration of this. This to me is the main strength of the paper, it is validating knowledge that is informally accepted in robotics in a systematic way.

The paper goes a step further though, and additionally proposes a novel method to fuse useful information from different frames to generate actions. It's very interesting (surprising, even) that a weighted combination of the proposed actions from each module in the network ends up producing a meaningful action that improves over using any one frame individually.

The other two FrameMiner methods beyond FM-MA are nice to show, but there is one more I think would be interesting to look at: take only the max-weighted action from the FM-MA architecture, i.e. don't do a linear combination but take the one with the highest weight. I suspect there may be diminishing contributions and that one frame ends up dominating in certain phases of different tasks, or that frames contribute equally enough that it won't matter which one is selected. I would gain more insight about the importance of the linear combination in FM-MA seeing this comparison.

The experiments are thorough and I appreciate the variety of tasks and platforms (mobile, bimanual, single-arm).

The limitations section is underwhelming. A big limitation that received no discussion is only PPO is looked at for behavior generation. It would be informative to look at other RL methods, but beyond RL, I think it would be much more informative to look also at imitation learning and model-based planning. As it stands the paper can primarily support the benefit of frame mining for RL, whether those benefits extend equally into other domains of policy learning and planning is unclear.

A final factor that I think is overlooked is the camera views. It is emphasized that acquiring new camera views is not necessary for the method and it only takes in a fused point cloud - however, the camera views still matter. For example, what if the camera is mounted on the robot's mobile base, or on the end-effector? Then the perception is situated to one of the frames identified as relevant for action. Does that change things w.r.t. to the conclusions the paper makes? I think discussing the aspect of a moving camera, if not showing additional experiments on it, is a worthwhile point to elaborate on.

**Summary Of Recommendation:**

I think this is a great paper, it had me excited while I was reading it, and I will advocate strongly for acceptance even if other reviewers feel differently. The results are simple, intuitive, and thorough. I believe the conclusions will have a strong impact on the manipulation learning community, since the reference frame one learns in is often an overlooked matter. If one can apply a simple method to fuse the best information from many reference frames without having to change the experimental setup at all (i.e. the cameras can stay wherever you want them), then it's an all around win. I do think there are more limitations to the method than discussed, which I describe in more detail in the strengths/weaknesses section. These are matters that can likely be addressed though in the rebuttal and in minor paper revisions.

---

> ### Author Response · Authors · 2022-08-25
> **Authors' Response for Reviewer vXBE**
>
> We sincerely thank you for your constructive feedback and your appreciation of our work! We address the comments and questions below:
>
> > Take only the max-weighted action from the FM-MA architecture.
>
> Please see “Response to Overlapping Reviewer Comments”.
>
>
> > It would be much more informative to look also at imitation learning and model-based planning.
>
> Please see “Response to Overlapping Reviewer Comments”.
>
>
> > A possibly moving camera that may be situated in an action-relevant frame (e.g. on the mobile base or on the end-effector).
>
>
> We would like to point out that our five tasks cover both static and moving camera settings (see Fig. 1). For the four tasks with moving cameras, a panoramic camera is mounted on the robot head.
>
> We agree camera placements can change the input point clouds being captured. Therefore, we add an ablation experiment in the supplementary (Section S.3.3) with cameras mounted on the robot base as suggested by the reviewer. Similar phenomena to our experiments in the main paper (Fig. 9) are still observed. Specifically, fusing multiple coordinate frames with our FrameMiners still leads to better sample complexity and final performance, demonstrating that FrameMiners are robust under different camera placements.
>
>
>
> > Typos and figure improvements.
>
> Thanks for pointing them out. We have already addressed them in our revision.

---

### Official Review · Reviewer_LU2t · 2022-08-01

**Originality:** Good
**Technical Quality:** Good
**Clarity Of Presentation:** Very Good
**Impact:** 3

**Recommendation:**

Weak Accept: I recommend accepting the paper, but will not argue for my recommendation if the majority of other reviewers have a different opinion.

**Summary:**

This paper studies the effect of different coordinate frames of point clouds on learning robotic manipulation skills with RL. 4 coordinate frames are investigated on 5 simulated manipulation tasks with 2 different 3D backbones. The experimental results demonstrate different coordinate frames indeed lead to different performances, and some coordinate frame, e.g., object-part frame, seems consistently better than other. Three schemes of fusing different coordinate frames are proposed and demonstrated to outperform single-frame baselines. The method is also demonstrated to work on a real-world object picking task.

**Issues:**

Please see the strenght/weakness sections.

**Quality Of The Limitations Section:**

Limitations are addressed clearly

**Reviewer Expertise:**

4: The reviewer is confident but not absolutely certain that the evaluation is correct

**Robotics Focus:**

Sufficient demonstration on hardware

**Strengths And Weaknesses:**

Strength:
- According to my best knowledge there has not been prior work investigating frame choice for 3D RL on robotic manipulation tasks, which seems to be an essential choice when doing RL with point clouds, so the problem itself is worth studying and thus the paper is well-motivated.
- The results are also interesting which demonstrates different performances with different coordinate frames. This suggests the tested 3D backbone networks are not se3 equivirant.
- The proposed FrameMiners method provides a free lunch to use for RL with point cloud inputs.

Weakness:
- The paper is fully empirical, with some limited discussion on why different frames lead to different performances. It would be nice if the authors can provide more insights into why different frames work so differently.
- If the only difference between different frames is just a se3 transformation, is it possible to learn a transformation layer to preprocess the input pointcloud before passing it to the policy? The transformation layer can be jointly trained using the RL loss. I would like to hear the author's thoughts on this.
- What frame is the robot action in? Is it always in a fixed frame, or does it change accoordingly when the input point cloud frame changes?
- There has been some recent work on transformation-invariant or transformation-equivirant point-cloud backbones / object representations [1][2]. The authors are encouraged to discuss if such works can be used to address the frame choice problem for robotic manipulation.
- The real-world experiments only demonstrate the proposed FM-MA policy can work. It would be good to know the baseline's performance on the real world as well to see if the difference also exists in the real world.

[1] Deng et al, Vector Neurons: A General Framework for SO(3)-Equivariant Networks
[2] Simeonov*, Du*, et al, Neural Descriptor Fields: SE(3)-Equivariant Object Representations for Manipulation


**Summary Of Recommendation:**

Please see the strenght/weakness sections.

---

> ### Author Response · Authors · 2022-08-25
> **Authors' Response for Reviewer LU2t**
>
> We sincerely thank you for your constructive feedback and your appreciation of our work! We address the comments and questions below.
>
> > Limited discussion on why different frames lead to different performances.
>
> Please see “Response to Overlapping Reviewer Comments”.
>
> > Is it possible to learn a transformation layer to preprocess the input point cloud before passing it to the policy?
>
> We have added an experiment in supplementary Section S.3.4. In this experiment, we add an additional network before the PointNet backbone to learn an adaptive $\mathbb{SE}(3)$ transformation based on the input point cloud. This transformation is then applied to the input point cloud before passing it through the PointNet backbone (note that we remove spatial transformation layers from the original PointNet in all of our experiments). However, we found that adding such a transformation prediction layer barely improves final performance.
>
> We conjecture that it's very difficult to predict a $\mathbb{SE}(3)$ transformation for aligning the input point cloud across time due to the weak supervision from RL training loss and the large search space (where most transformations are ineffective). Moreover, in many challenging tasks, we may need to fuse information simultaneously from multiple coordinate frames (e.g., left-hand and right-hand frames). This is not achievable through learning a single transformation. In contrast, our FrameMiners take advantage of easily-accessible frame information (e.g., end-effector poses) to align the point clouds without relying on transformation prediction. We then fuse the merits of multiple candidate coordinate frames.
>
> > What frame is the robot action in?
>
> The x,y velocities and angular velocity of the moving platform (i.e., robot base) are represented in the robot-base frame. Other action components (i.e., robot joint velocities and finger positions of end-effectors) do not have a frame, because they represent the relative movement of two robot links. We also add an experiment where we represent both actions for the robot base and input point clouds in the world frame. We did not observe performance improvements over using world-frame point clouds and base-frame actions.
>
>
> > Recent work on transformation-invariant or transformation-equivariant point-cloud backbones.
>
> While $\mathbb{SO}(3)$ and $\mathbb{SE}(3)$-equivariant/invariant point-cloud backbones are of great benefit for analysis within each object (e.g., shape classification, part segmentation, and 6D pose estimation), our robot-object interaction setting is a bit different.
>
> In robot manipulation scenarios, a particular challenge comes from inferring the relations between two object parts (e.g., relative pose between the end-effector and the cabinet handle). This binary relation inference task is challenging under the weak RL loss supervision, even using $\mathbb{SO}(3)$ and $\mathbb{SE}(3)$ equivariant/invariant backbones. FrameMiners explicitly approach this challenge by aligning point clouds (across multiple time steps) with the known transformation matrices (e.g., the end-effector pose). This reduces many binary relation inference tasks to single-subject location tasks, which have much lower difficulty. For example, when using the end-effector frame in the OpenCabinet task, the network only needs to copy the handle pose to infer the relative pose between the handle and the end-effector, as the end-effector is always at the frame origin (see Fig. 5).
>
> We have added this discussion in the supplementary (Section S.3.5) and cited the related works on $\mathbb{SO}(3)$ and $\mathbb{SE}(3)$ equivariant/invariant networks.
>
>
> > Whether differences between baseline and FM-MA also exists in the real world.
>
> We have updated our real experiment section. The answer is affirmative. We use the model trained with 4M samples in simulation for deployment in the real world. Using FrameMiner-MixAction (FM-MA), we achieve an 84% success rate on PickObject. The end-effector frame achieves an 80% success rate. The robot-base frame is unable to achieve successful picks under our training budget. Note that the differences in the real world are very similar to the simulation environment (as in Fig. 4 & 10). The results indicate that frame choice or mining does not introduce an extra domain gap, and FM-MA is an effective strategy for real robots.

---

### Official Review · Reviewer_1Hcu · 2022-08-02

**Originality:** Very Good
**Technical Quality:** Good
**Clarity Of Presentation:** Very Good
**Impact:** 4

**Recommendation:**

Weak Accept: I recommend accepting the paper, but will not argue for my recommendation if the majority of other reviewers have a different opinion.

**Summary:**

This paper presents an approach for fusing different coordinate frames for point clouds in robot manipulation tasks. In robot manipulation, you can use different coordinate frames to express the given point cloud. The paper considers five different coordinate frames: the world frame, which is a fixed frame, robot base frame, which is a frame attached to the robot's base, an end-effector frame, which is attached to the robot's end-effector, and target-part frame, which is attached to the part of the target object the robot wishes to manipulate. The paper demonstrates the impact that each coordinate frame has on learning in five different manipulation tasks, and propose an architecture that can fuse different frames to produce an action.

**Issues:**

Please see the summary of the recommendations above.

**Quality Of The Limitations Section:**

Additional details required

**Reviewer Expertise:**

4: The reviewer is confident but not absolutely certain that the evaluation is correct

**Robotics Focus:**

Sufficient demonstration on hardware

**Strengths And Weaknesses:**

Strengths:
- Delves into the important-yet-often-overlooked problem of choosing a frame for sensory data in robot manipulation
- Performs thorough empirical analysis of their hypotheses (ex. different frames lead to different performances; a fusion of point clouds expressed in different frames will lead to better performance)

Weaknesses:
- The paper lacks a detailed analysis of why they have chosen a particular manipulation task to validate their hypotheses. Currently, it sounds like they have been selected just because they were available.
- A more thorough analysis of empirical results is needed.

**Summary Of Recommendation:**

I think this is a very helpful paper in general, but it could use additional analyses on the empirical results. More concretely, why did you choose the five tasks? Was it because they were available? It seems that you have chosen them based on their merits - ex. some require movement of the base, some don't; some require moving a part of an object and some don't, etc. It would be beneficial for the reader if you could provide why these particular tasks have been chosen so that they can clearly understand what you are trying to test in each task.

Second, I think you need a more elaborated and thorough description of your insights and experimental results. For instance, I found lines 170-175, which outline your insights as to why certain frames would help to learn, to be very interesting and helpful, and would have liked to see more validation of these insights than the ones you provided in lines 201-202. Also, I would like to know why Table 1 and Figure 9 are the way they are. Is it just the fusion of the point cloud expressed in different frames that result in this? Or was there something else? If it is the former, why is that so? If it was the latter, what were the other factors?

Lastly, for frames such as target-object-part frame, it is important where on the object you choose your frame (ex. you chose the handle). So, it seems that human judgment is not only needed in choosing the type of frame but also the where on the object. The paper needs to be more upfront about this.

---

> ### Author Response · Authors · 2022-08-25
> **Authors' Response for Reviewer 1Hcu**
>
> We sincerely thank you for your constructive feedback and appreciation of our work! We address the comments and questions below.
>
> > More discussions / motivations on why particular tasks are chosen.
>
> We aim to cover a wide range of factors that may influence the selection of point cloud coordinate frames. Specifically, the tasks are chosen to cover various robot mobilities, numbers of robot arms, and camera placements, as demonstrated in Figure 1.
>
> Different robot mobility results in differences in the world frame and the robot-base frame. These two frames are aligned in static robots but not in mobile robots. The robot's mobility can also change the focus of tasks (e.g., navigation or object interaction), which may place different requirements on the choice of point cloud frame.
>
> We cover both single-arm and dual-arm environments, as they pose different requirements for point cloud frame selection. In single-arm environments, using the only end-effector frame may already be able to achieve good performance. However, in dual-arm environments, there are two end-effector frames, and these tasks require precise coordination between the two robot arms, which poses significant challenges for manipulation learning. As each end-effector may have a preferred frame, the necessity of frame fusion becomes more pronounced.
>
> Last but not least, camera placements determine sources of point clouds, potentially influencing the selection of coordinate frames. In our experiments, we cover both static and moving camera placements (mounted on robots).
>
> Due to page limits, we have included this discussion in Section S.4 of the supplementary.
>
>
> > More elaborated and thorough description of your insights and experimental results; Why Table 1 and Figure 9 are the way they are. Is it just the fusion of the point cloud expressed in different frames that result in this? Or was there something else?
>
> Yes, it’s the fusion of multiple coordinate frames that causes the outperformance of FrameMiners, not other factors. Please see “Response to Overlapping Reviewer Comments” for a detailed explanation.
>
>
> > For target object-part frame, human judgment is not only needed in choosing the type of frame but also the where on the object.
>
> We have added this discussion in the limitation section. We would also like to point out that, while humans need to propose target object-part frame *candidates* for each object category, our FrameMiners can automatically select the best suitable one from them or fuse the merits of multiple candidates. We thus don’t need many human priors or experimental trials to select the right coordinate frame, and we can simply include all major object parts as candidates.

---

### Official Review · Reviewer_aJn6 · 2022-08-02

**Originality:** Good
**Technical Quality:** Very Good
**Clarity Of Presentation:** Very Good
**Impact:** 4

**Recommendation:**

Strong Accept: I recommend accepting the paper and will argue for my recommendation even if other reviewers hold a different opinion.

**Summary:**

The paper is motivated by the fact that different coordinate frames have a significant impact for point cloud-based reinforcement learning for manipulation tasks. In a set of manipualtion tasks involving mobile and dual-arm manipulators, the paper compares with four coordinate frames: World frame, robot-base frame, end-effector frame and target-part frame and show that not a single frame can consistenly outperform the reset. The paper then proposes multiple ways of combining different coordinate frames by incoporating point clouds in different frames into the input and use the RL loss to find a combination of different frames. The proposed frame combination method achieves significant performance gain over single-frame baselines.

**Issues:**

Please see my weakness in the above sections.

For the robotics focus, I chooose "no hardware experiments" as the current hardware experiments are not sufficient to demonstrate the core idea of the proposed method.



**Quality Of The Limitations Section:**

Limitations are addressed clearly

**Reviewer Expertise:**

5: The reviewer is absolutely certain that the evaluation is correct and very familiar with the relevant literature

**Robotics Focus:**

Highly relevant to robotics but no hardware experiments

**Strengths And Weaknesses:**

### Strengths
1. The paper is well motivated and can bring the community's attention to the importance of choosing the right coordinate frames, which has been so far less explored.
2. The paper is well structured and easy to follow. The tables, figures, their captions as well as the video in the supplement are very informative.
3. The arguments in the paper are well supported by the quantitative and qualitative resutls.
4. Finally, the proposed method is reasonable and well motivated. The supporting experiments in Figure 8 show interesting insights into how the method works.

### Weakness
The weaknesses I listed below are all minor and serve mostly as suggestions to improve on the paper.
1. The real world experiment is underwhelming.
* Due to the sample complexity of reinforcement learning, there is no training in the real world and only the trained policy is transferred. In this case, the sample complexity in the simulation becomes less of a concern. As such, I wonder if with more training samples, do different choices of coordinate frames yield similar asymptotic performance?
* Only the pick object is done in the real world. For this case, the end-effector frame is already achieving the best performance. It would be very helpful to demonstrate one more task in the real world where frameminer outperforms single-frame baselines.
* From the visualization in the paper, it seems that the full point cloud is used in simulation. How does this transfer to the real world?
2. In the current method, actions from different coordiante frames are averaged. This seems to be a strange choice to me. That means all coordinate frames always contribute to the final actions. Why not sample action from one of the frames based on the weight?
3. Only RL setting is demonstrated in this case. What about imitation learning? It seems that the same principle should apply as well. This could further enlarge the impact of this work. A related work in this area is Bowen et al. 2022 [1].

[1] Wen, Bowen, et al. "You Only Demonstrate Once: Category-Level Manipulation from Single Visual Demonstration." RSS 2022.

**Summary Of Recommendation:**

Overall, the paper presents interesting finds with sufficient supporting experiments that deserve publication at the conference. I believe these findings will be beneficial for other researchers in the community.

---

> ### Author Response · Authors · 2022-08-25
> **Authors' Response for Reviewer aJn6**
>
> We sincerely thank you for your constructive feedback and your appreciation of our work! We address the comments and questions below:
>
> > From the visualization in the paper, it seems that the full point cloud is used in simulation.
>
> All of our simulation experiments use *partial and occluded*  point clouds captured by cameras, which aligns with the real-world setting. In Figure 3, we visualize point clouds of an OpenCabinetDoor trajectory, which are captured by a panoramic camera mounted on the robot head. When we perform manipulation learning, the input point clouds are rather sparse, which can cause difficulties in shape visualization. For better visualization, we show dense point clouds in Fig 3, which is achieved by rasterizing point clouds with higher resolution in our simulation. We have clarified this in the caption of Figure 3.
>
>
> > The sample complexity becomes less of a concern in simulation.
>
> We would like to point out that, even in simulation, sample complexity is still important, as the training can be rather time-consuming. Specifically, while our method has a lower sample complexity, it still typically takes 3-4 days to converge in simulation. This is mainly because our task setup realistically reflects the difficulty of manipulation skill learning by using a diverse set of 3D objects with complex topology and geometry.
>
> > If with more training samples, do different choices of coordinate frames yield similar asymptotic performance?
>
> We find that even if a large number of training samples are given, final performances still vary significantly (see Figure 9 and Table 1). Without a good point cloud frame selection or fusion of multiple frames, policy learning could be much harder, and RL algorithms may fail to find a successful policy.
>
> > Only the pick object is done in the real world; it would be helpful to demonstrate one more task in the real world where FrameMiner outperforms single-frame baselines.
>
> Yes, it would be more convincing to see real-world experiments on other tasks as well. We have started working on a new real-world task (cabinet door opening with a fixed arm). Due to the tight deadline of the rebuttal period, we will add this experiment in camera-ready.
>
> > In the current method, actions from different coordinate frames are averaged.
>
> Please see “Response to Overlapping Reviewer Comments”.
>
> > Imitation learning in addition to RL.
>
> Please see “Response to Overlapping Reviewer Comments”. We have also cited the related work.

---

### Author Response · Authors · 2022-08-25
**Response to Overlapping Reviewer Comments (2/2)**

>   Why different frames lead to different performances; more elaborated and thorough description of your insights and experimental results **[Reviewer 1Hcu, LU2t]**

In Figure 3, we analyze how different coordinate frames canonicalize input point clouds in different manners, which essentially perform various alignments among point clouds across multiple time steps (L170-175). For example, by representing input point clouds in the end-effector frame, the end-effector is always aligned at the origin throughout a trajectory, such that the visual network does not need to locate the end-effector from point clouds. Similarly, the robot-base frame naturally aligns its frame axes with the moving directions of the robot's base, which may also simplify the learning of visual modules.

In robot manipulation tasks, a particular challenge comes from inferring binary relations between object parts (e.g., relative pose between the end-effector and the cabinet handle). However, by aligning point clouds under certain frames (e.g., the end-effector frame), these binary relation inference tasks may be reduced to single-subject location tasks (e.g., simply copying the handle pose), which become much easier to solve. To verify this point, we perform a diagnostic experiment on OpenCabinetDoor, where we intentionally remove all robot points. As shown in Figure 5, after the robot points are removed, the performance of the end-effector frame remains the same, while the performance of the robot-base frame drops a lot. This confirms that aligning point clouds in the end-effector frame significantly reduces the difficulty of relation inference between the robot end-effector and the cabinet handle. We have added this discussion in Section 3.4.

We demonstrate the effectiveness of FrameMiners in Table 1 and Figure 9. We believe this is mainly because different action components (e.g., actions for the left hand, right hand, and robot base) and task skills/manipulation stages (e.g., navigation and precise object manipulation) are often suited for different coordinate frames. Therefore, combining merits from different coordinate frames simplify visual module learning and yields better sample efficiency and well-performing policies. To demonstrate this point, we visualize the learned weights of FrameMiner-MixAction (FM-MA) over a MoveBucket trajectory in Figure 8. We provide our analysis in Line 253-268, and show that (1) different robot joints have distinct frame preferences; (2) the weight contribution from each frame changes greatly over time across different trajectory stages (i.e., approaching the bucket, moving the bucket to the platform, and placing the bucket on the platform). We added a forward reference in Section 4.2 to make the connection between analysis and results more clear.

---

### Author Response · Authors · 2022-08-25
**Response to Overlapping Reviewer Comments (1/2)**

We sincerely thank Reviewers *aJn6, 1Hcu, LU2t, vXBE* and the AC for your constructive feedback and your appreciation of our work! We address overlapping comments in this post and other comments in the responses to each reviewer.

> Imitation learning in addition to RL **[Reviewer aJn6, vXBE]**

We have added an experiment in the supplementary (Section S.3.1) to demonstrate that our findings can generalize to manipulation learning algorithms of other domains besides online RL. In this experiment, we perform behavior cloning (BC) by representing input point clouds under different coordinate  frames, along with using our proposed FrameMiner-MixAction (FM-MA). We observe similar findings to the original online RL experiments. Specifically, point cloud frame selections have a profound impact on BC’s performance. Our proposed FrameMiner is capable of automatically selecting the best single frame from multiple frame candidates or combining the merits from them and outperforming single-frame baselines.


> Take only the max-weighted action instead of weight linear combination **[Reviewer aJn6, vXBE]**

We denote our original FrameMiner-MixAction that uses weighted linear combination to fuse action proposals from each coordinate frame (see Figure 7) as FM-MA-WLC. We denote the alternative design that chooses the max-weighted action proposal for each joint as FM-MA-MW. Note that FM-MA-WLC uses SoftMax to normalize the weights; thus, FM-MA-WLC can be regarded as a “soft version'' of FM-MA-MW.

We agree that the “max-weight” strategy is more intuitive, so we added two experiments in the supplementary (Section S.3.2):
- We train FM-MA-MW from scratch.
- We resume from the final checkpoint of the original FM-MA-WLC. During evaluation, we use the max-weighted action proposal (MW) as the action output.

We observe that for both experiments, using FM-MA-MW deteriorates performance. We conjecture that FM-MA-WLC alleviates optimization difficulty, which likely comes from the fact that FM-MA-WLC is a “soft-version'' of FM-MA-MW with well-behaving gradients. On the other hand, since FM-MA-MW uses argmax operation, there is a lack of gradient during training, which leads to more difficult optimization.

---

### Author Response · Authors · 2022-08-25
**Updated Paper and Supplementary**

Dear Reviewers and ACs,

Thank you for your constructive feedback! We have updated our paper and supplementary (PDF attached). We have used red color to indicate our changes.

---

### Meta-Review · Area_Chair_FQ85 · 2022-08-06

**Recommendation:** Accept (Poster)
**Confidence:** 5

**Metareview:**

There is consistent agreement from all 4 reviewers that this is a well structured paper with contributions that can potentially provide broad impact in robotics and are well supported by results and evidence.

**Best Paper Nomination:**

No

---

> ### Author Response · Authors · 2022-08-25
> **Authors' Response for Area Chair FQ85 (2/2)**
>
> > The chosen manipulation tasks are not well motivated.
>
> We have addressed this by providing detailed motivations of our task choice in the response to Reviewer 1Hcu, and include these motivations in the supplementary (Section S.4). We would like to also note that, in this work, we choose tasks to cover a wide range of factors that influence coordinate frame selection, such as robot mobilities, numbers of robot arms, and camera placements, as demonstrated in Figure 1.
>
>
>
> > A more thorough analysis and discussion of empirical results.
>
> We have addressed this in “Response to Overlapping Reviewer Comments”. We have included a more elaborated discussion in Section 3.4 and added a forward reference in Section 4.2 to make the connection between analysis and results more clear.
>
>
>
> > Alternative methods beyond averaging actions from all coordinate frames
>
> We have addressed this in “Response to Overlapping Reviewer Comments”. We followed the reviewer’s suggestions and added experiments in the supplementary (Section S.3.2) to compare the two designs. We observe that our original design performs better.
>
>
> > Relationships to recent transformation-invariant or transformation-equivariant methods
>
> We have addressed this in our response to Reviewer LU2t and added discussions in the supplementary (Section S.3.5).
>
>
> > The importance of camera views should not be downplayed
>
> We have addressed this in our response to Reviewer vXBE, and we added experiments in the supplementary (Section S.3.3). When changing the camera placement as suggested by Reviewer vXBE, we observe similar phenomena as in the original experiments. We show that FrameMiners’ effectiveness is robust under different camera configurations.
>
> > Other behavior generation methods are of interest, such as IL or model-based methods.
>
> We have addressed this in “Response to Overlapping Reviewer Comments”, and we added experiments on behavior cloning (BC) in the supplementary (Section S.3.1). Similar to online RL, we observe that point cloud frame choice has a profound impact on BC’s performance, and that our FrameMiner can benefit manipulation learning. These results demonstrate that our contributions can benefit learning algorithms of other domains besides online RL.

---

> ### Author Response · Authors · 2022-08-25
> **Authors' Response for Area Chair FQ85 (1/2)**
>
> Thank you for your constructive feedback! We would like to point out that all four reviewers (aJn6, 1Hcu, LU2t, vXBE) are enthusiastic and supportive of our work, thinking our paper is “well-motivated” (Reviewer LU2t,aJn6), our results are “simple, intuitive, and thorough” (Reviewer vXBE), with “interesting insights” (Reviewer aJn6) and “thorough empirical analysis of hypotheses” (Reviewer 1Hcu). Here is a summary of how we have addressed each point of weakness:
>
>
> > The real-world experiment is minor and not well investigated. Many baselines are missing, which puts the findings in jeopardy, as they may only hold up in simulation.
>
> 1. In most RL works, the real world is where policies are deployed, and the simulation environment is where policies are trained. Therefore, we evaluate training performance in simulation and deployment performance in the real world. The real-world experiment is designed to show that our trained policy can be deployed without introducing extra domain gaps, which will be discussed in detail in Bullet 4 & 5 with new real-world baseline results. We would like to point out that the focus of this paper is to study how we can reduce the sample complexity at training time, which happens in simulators, because training takes a long time for realistic manipulation tasks (see Bullet 2). For this research focus, our main finding is that *selecting and mining frames* for point cloud representation has a profound impact. We made sufficient comparisons with **baseline** RL methods, which are single-frame methods. In fact, Reviewer 1Hcu and Reviewer vXBE said that our experiments are “thorough”.
>
> 2. In principle, one can directly train RL in the real world with our method. However, we want to emphasize that this will be extremely costly – even in simulation, it takes days to train. While our approach has lower sample complexity, it still typically takes 3-4 days to converge in simulation. This is mainly because our task setup realistically reflects the difficulty of manipulation skill learning by using a diverse set of 3D objects with complex topology and geometry.
>
> 3. Although there aren’t many results about the choice of point cloud coordinate frames in the emerging field of 3D RL, frame selection (e.g., camera view) is fundamental in robotics. As Reviewer vXBE said, “I believe the claims being made regarding the benefits of particular frames benefitting different tasks is known in robotics, but I have not seen a rigorous demonstration of this”. Besides robotics, the importance of selecting coordinate frames for point clouds is also reported in the 3D vision community (e.g., [1,2] show that frame transformation is important for 3D detection on the real-world KITTI dataset). In fact, all reviewers agree that our method is “well-motivated”, and we provide “interesting insights” and “thorough empirical analysis of hypotheses”.
>
> 4. When deploying our trained policy in the real world, the domain gap will primarily come from two sources: 1) frame localization (e.g., the robot-base frame and the end-effector frame) in the real world, and 2) visual perception from point cloud networks. The success of our PickObject experiment in the real world shows that these two factors are accounted for. This is because 1) accurate localization of many frames is available through robot-proprioceptive information, and 2) the simulator used in our experiments (SAPIEN) has a low sim2real domain gap by depth sensor simulation and full-physics contact modeling.
>
> 5. Regarding *missing baselines*, the request is from Reviewer LU2t, who pointed out that the single-frame baseline's performance in the real world is missing, and would like to know if the differences also exist in the real world. We have updated our real experiment section, and the answer is affirmative. We find that the differences between various methods in the real world are very similar to the simulation environment (as in Fig. 4 & 10). This indicates that frame choice or mining does not cause an additional domain gap, and FM-MA is an effective strategy for real robots.
>
> [1] Qi, Charles R., et al. "Frustum PointNets for 3D Object Detection from RGB-D data"
>
> [2] Zhou, Yin, et al. "End-to-End Multi-View Fusion for 3D Object Detection in LiDAR Point Clouds."